# Immune modules to guide diagnosis and personalized treatment of inflammatory skin diseases

Teofila Seremet [1,6], Jeremy Di Domizio [1,6], Antoine Girardin[1,6], Ahmad Yatim[1], Raphael Jenelten [1], Francesco Messina[1], Fanny Saidoune[1], Christoph Schlapbach [2], Sofia Bogiatzi [1], Frederic Minisini[1], Natalie Garzorz-Stark[3], Matthieu Leuenberger[1], Héloise Wüthrich[1], Maxime Vernez[1], Daniel Hohl[1], Stefanie Eyerich [3], Kilian Eyerich [3], Emmanuella Guenova [1], Carle Paul[4], Raphael Gottardo [5], Curdin Conrad [1] & Michel Gilliet [1] ✉

Previous advances have identified immune pathways associated with inflammatory skin diseases, leading to the development of targeted therapies. However, there is a lack of molecular approaches that delineate these pathways at the individual patient level for personalized diagnostic and therapeutic guidance. Here, we conduct a cross-comparison of expression profiles from multiple inflammatory skin diseases to identify gene modules defining relevant immune pathways. Seven modules are identified, representing key immune pathways: Th17, Th2, Th1, Type I IFNs, neutrophilic, macrophagic, and eosinophilic. These modules allow the development of a molecular map with high diagnostic efficacy for inflammatory skin diseases and clinico-pathologically undetermined cases. Aligning dominant modules with treatment targets offers a rational framework for treatment selection, improving response rates in both treatment-naïve patients and non-responders to targeted therapies. Overall, our approach offers precision medicine for inflammatory skin diseases, utilizing transcriptional modules to support diagnosis and guide personalized treatment selection.

Over the last decade, dermatology has undergone a remarkable translational revolution marked by the emergence of numerous molecular-based therapies for inflammatory skin conditions. This transformative progress owes its success to a profound comprehension of immune pathways, encompassing both adaptive pathways linked to T helper (Th) cell differentiation Th1, Th2[1], Th17[2], T regulatory cells[3] and innate pathways associated with the generation of type I IFNs[4] and IL-1 cytokines[5]. These pathways are now established as integral to the pathogenesis of common inflammatory skin disorders, such as psoriasis, characterized by Th17 involvement[6–8], atopic dermatitis (AD), characterized by Th2 involvement[9–11], lichen planus (LP), characterized by Th1 involvement[12], lupus erythematosus (LE), a type I IFN-related disorder[13,14], and neutrophilic diseases, influenced by cytokines of the IL-1 family[15]. Molecular-based therapies have emerged

¹Department of Dermatology, Lausanne University Hospital CHUV and University of Lausanne, 1011 Lausanne, Switzerland. ²Department of Dermatology, Inselspital, Bern University Hospital, University of Bern, Bern, Switzerland. ³Department of Dermatology and Venereology, Medical Center, University of Freiburg, Freiburg, Germany. ⁴Department of Dermatology and Venereology, CHU Toulouse, Toulouse, France. ⁵Biomedical Data Science Center, CHUV, UNIL, and SIB, Lausanne, Switzerland. ⁶These authors contributed equally: Teofila Seremet, Jeremy Di Domizio, Antoine Girardin. ✉e-mail: michel.gilliet@chuv.ch

as effective means to target these pathways, including anti-IL23 and IL-17A/F monoclonal antibodies for Th17 inhibition in psoriasis[16–21], anti-IL-4RA and anti-IL-13 monoclonal antibodies for Th2 blockade in AD[22–24], JAK1/2 inhibitors for Th1 inhibition in LP[12,25,26], anti-IFN-αβ receptor (IFNAR) monoclonal antibodies for type I IFN signalling blockage in LE[27], and anti-IL-1R and anti-IL-36R to mitigate neutrophilic inflammation in neutrophilic diseases[28–33]. Despite the substantial therapeutic efficacy of these molecular treatments, clinicians often encounter patients who do not respond, prompting questions about the accuracy of diagnosis, appropriateness of the chosen therapeutic target, or the possibility of an immune shift under immune pathway blockade.

Presently, a notable gap exists in molecular approaches capable of delineating the relevant immune pathways at the individual patient level. Most studies of skin diseases have either concentrated on single diseases and their controls or attempted to identify single biomarkers through comparisons between multiple skin diseases[34,35], without providing an answer to the above questions applicable to the routine clinical practice.

In the current study we establish a systematic molecular cartography, grounded in functionally relevant immune modules, to be used for the diagnosis of inflammatory skin diseases in the routine clinical setting. This module-based map is superior in diagnosing challenging cases like erythrodermas or undetermined rashes, compared to existing clinico-pathological standards. Furthermore, matching the dominant module with the treatment target offers a logical framework for treatment selection, benefitting both treatment-naïve patients and those who do not respond to targeted therapies.

## Results

### Transcriptional cross-comparison of inflammatory skin diseases identifies core immune modules

To create a molecular immune map of inflammatory skin diseases, we focused on skin conditions known to involve specific immune pathways and to respond to treatments targeting these pathways. These included psoriasis ($n = 25$), AD ($n = 17$), LP ($n = 12$), cutaneous LE ($n = 12$), neutrophilic diseases ($n = 10$), such as pyoderma gangrenosum, dissecting cellulitis, and hidradenitis suppurativa, which involve Th17, Th2, Th1, type I IFN and neutrophil-mediated inflammation, respectively. Biopsies of patients with clinically and histologically well-defined diseases were selected and referred to as "sentinels". Patient selection criteria are detailed in Material and Methods, and representative clinical and histological images are provided in Suppl. Figure 1. We employed NanoString transcriptomics profiling to examine the expression of 600 immune-related genes in these sentinel biopsies. Visualizing the data using Uniform Manifold Approximation and Projection (UMAP) revealed a certain degree of clustering of samples based on disease type (Suppl. Fig. 2), indicating similarities in gene expression patterns among individuals with the same condition. To elucidate the specific immune pathways involved in disease clustering, we conducted differential gene expression analysis for each sentinel compared to all other diseases (Fig. 1a). We found that psoriasis samples exhibited differential expression of genes associated with Th17 inflammation, including the Th17 recruiting chemokine CCL20; NOS2 and ARG1/2, which stimulate Th17 differentiation; the Th17 cytokines IL-17A and IL-17F; NFKBIZ, a mediator of Th17 signalling in keratinocytes; the Th17-induced antimicrobial peptides DEFB4A, DEFB103B, S100A9 and S100A8; the Th17-induced cytokines IL-19, IL-36A, IL-36G, IL-36RN, and CXCL8; and the Th17-dependent pro-angiogenic factor CXCR2 (Fig. 1a). AD samples showed differential expression of genes linked to Th2 inflammation, including various chemokines attracting Th2 cells and eosinophils (CCL22, CCL26, CCL13, CCL18) (Fig. 1a). LP samples exhibited differential expression of genes related to Th1-type inflammation, including cytokines like IL-12B; Th1 co-stimulators CD30 and CD27; molecules involved in Th1

differentiation (TNARSF9) or Th1/Tc1 cell cytotoxicity (KLRC4); Th1-induced MHC molecules HLA-DP1 and HLA-DPA1; the anti-apoptotic molecule TRAF1; and genes involved in T regulatory cell functions (FOXP3, ICOS, CTLA4, DUSP4) (Fig. 1a). Cutaneous LE samples displayed differential expression of genes associated with type I IFN-mediated inflammation, such as TLR3, MYD88, IRF7, PSMB9, (involved in the induction of IFN expression); STAT2 and PRF1 (involved in IFNAR signalling); or IFN-stimulated genes CXCL11, CXCL9, CCL8, CCL5, ISG15, IFIT2, IFITM1, IFI35, IFIH1, Mx11, BLYS, BST2, LAG3 (Fig. 1a). Neutrophilic diseases showed differential expression of genes related to myeloid cells. To dissect the myeloid signature, we compared neutrophil diseases, containing high numbers of neutrophils, with skin manifestations of severe COVID that are strongly infiltrated by macrophages[36]. We also added Wells disease, known as eosinophilic cellulitis, a rare inflammatory condition characterized by pathogenic eosinophil infiltration of the dermis. By comparing these diseases, the myeloid signature was subdivided into gene groups related to neutrophils, macrophages, and eosinophiles (Suppl. Fig. 3). Neutrophilic diseases predominantly expressed genes related to neutrophils with expressions of PECAM1 (receptor involved in neutrophil transmigration), IL-1B and IL-6 (regulator of neutrophil trafficking), CR1, TNFAIP6, MME (neutrophil receptors and molecules), RUNX1, SPP1 (neutrophil attracting chemokine), neutrophil-secreted chemokines (CCL3, CCL4). Wells syndrome samples showed a specific eosinophil signature (PPBP, CD59, leukemia inhibiting factor LIF) or related to eosinophil recruitment (BATF3). COVID skin manifestation samples displayed a dominant macrophage signature with expression of macrophage receptors (CD14, CSF3R, FCGR2A, FCGR3A/B CLEC4A, CD163, CD209, MRC1, CD40), and differentiation factors (CSF1) as previously reported[36]. In summary, our cross-comparative immune profiling of inflammatory skin diseases identified seven gene groups defining the following functional immune modules: Th1, Th17, Th2, type I IFN, neutrophilic, macrophage-associated, and eosinophilic.

### Module profiles efficiently classify inflammatory skin diseases

Having identified functional immune modules underlying model inflammatory skin diseases, we investigated their potential utility as diagnostic markers for these conditions. Module genes were found to exhibit remarkable efficacy in the clustering of disease samples with concordant diagnoses, as evidenced by both UMAP (Fig. 1b) and heatmap analyses (Fig. 1c). The clustering efficiency per disease, assessed via the Fowlkes-Mallows (FM) index, markedly improved when utilizing module genes (FM index of 0.95) compared to either the complete gene panel (FM index of 0.74) or a minimal gene set obtained through NS Forest (FM index of 0.84) (Fig. 1d). Remarkably, module genes also facilitated disease clustering of RNAseq transcriptomics data more effectively than the 20,000 genes (FM index of 0.89 for module genes versus FM index of 0.60 for all genes) (Suppl. Figure 4), albeit not reaching the clustering efficiency of the data generated by Nanostring. Together these observations suggest that immune modules represent a functionally relevant set of genes capable of disease classification across experimental platforms.

Heatmap visualization revealed that clustering was associated with the expression of a dominant module (predominantly Th17, Th2, Th1, I-IFN, neutrophilic and eosinophilic for diseases psoriasis, AD, LP CLE, neutrophilic diseases, and Wells syndrome, respectively) while other modules exhibited either absence or lower expression levels (Fig. 1c). Computation of module scores, defined as the mean expression levels of all genes within the module, allowed us to establish module dominance criteria: an expression level surpassing a threshold of at least 0.5 in the normalized plot and being significantly greater than all other modules (Fig. 2a, b and Suppl. Table 1). Based on this definition, we observed a strong association between the dominant module and the disease diagnosis (Fig. 2b), leading to a strong classification accuracy (Table 1). Furthermore, validation using an

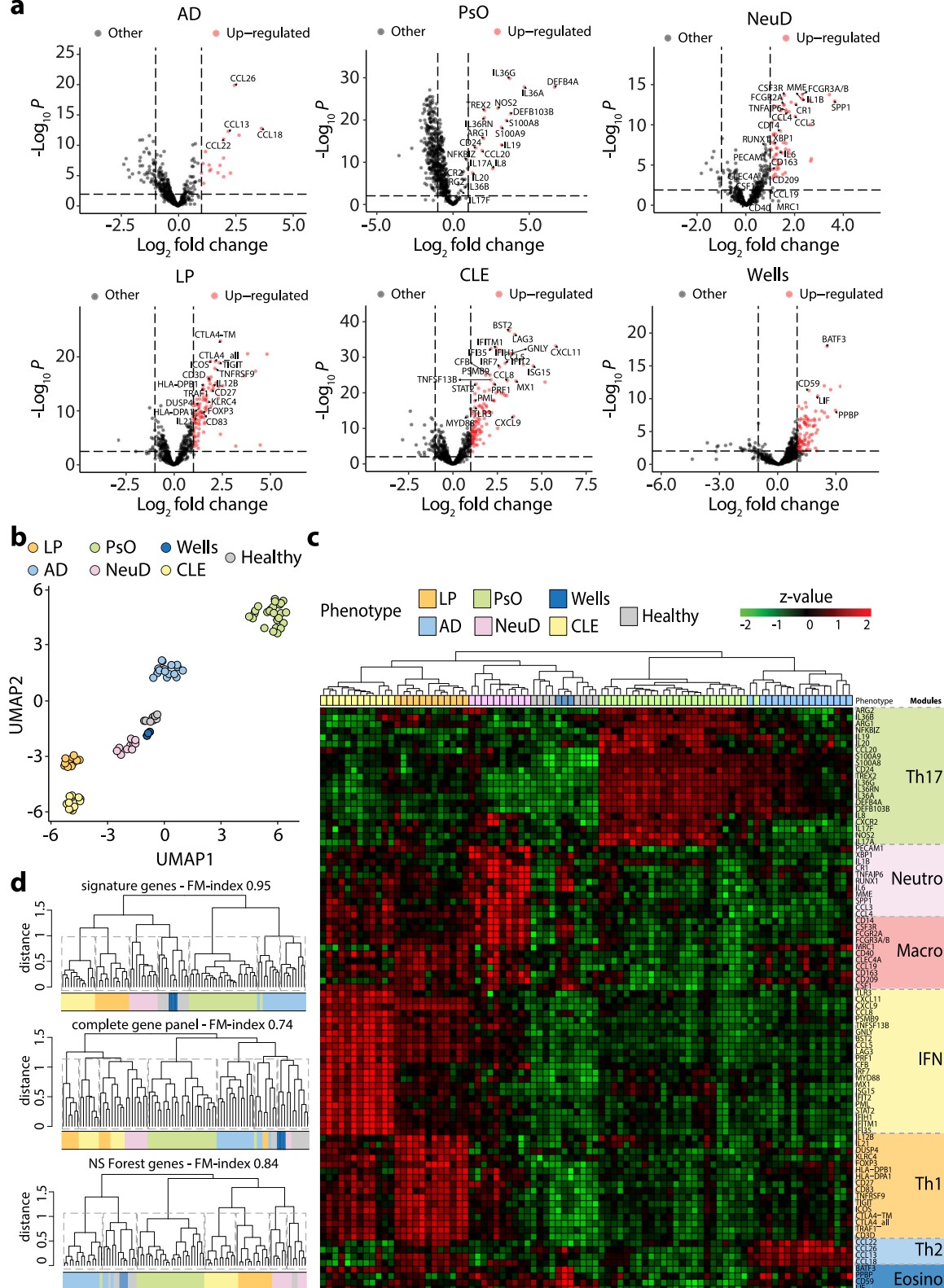

**Fig. 1 | Identification of immune modules through cross-comparison of inflammatory skin disease profiles. a** Sentinel biopsies from psoriasis (PsO), atopic dermatitis (AD), neutrophilic diseases (NeuD), lichen planus (LP), cutaneous lupus erythematosus (CLE), and Wells syndrome (Wells) patients were profiled. The expression levels for each disease were plotted against profiles of all other diseases to identify differentially expressed genes (DEG) that define modules. Volcano plots depict DEGs, with dashed lines indicating the significance threshold defined as log2 fold change > 2 and *p* value < 0.01 derived from two-sided t-tests. **b** UMAP projection of sentinel profiles based on module expression, demonstrating clustering according to disease. **c** Heatmap of sentinel profiles based on modules, showing hierarchical clustering along with disease-specific expression of modules. The color gradient reflects expression levels given as z-scores. **d** Hierarchical clustering of sentinel profiles based on expression of either module genes, the complete gene panel, or NS Forest minimal gene markers, shown by dendrograms. Clustering accuracy is indicated by the Fowlkes–Mallows (FM) index.

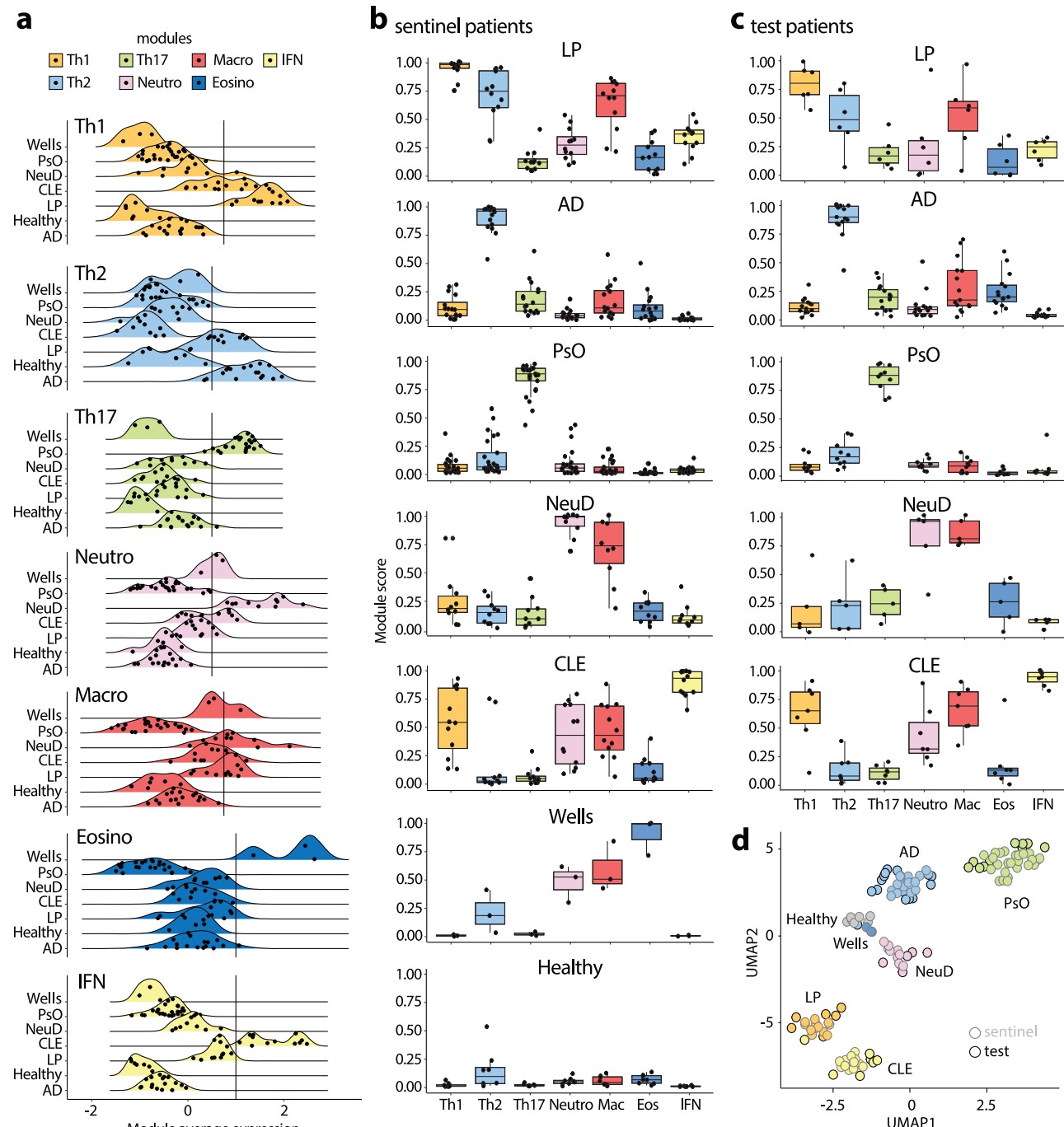

**Fig. 2 | Module-based profiles can classify inflammatory skin diseases.**
**a** Ridgeline plots depict Th1, Th2, Th17, type I interferon (IFN), neutrophilic (neutro), macrophagic (macro) and eosinophilic (eosino) module scores in sentinel biopsies across model inflammatory skin diseases (PsO, AD, LP, CLE, NeuD, Wells). Solid lines indicate the thresholds of module activation. **b** Box plots showing the normalized modules scores in sentinel biopsies across all model diseases. Each dot represents one sentinel biopsy. LP ($n = 12$), AD ($n = 16$), PsO ($n = 25$), NeuD ($n = 10$), CLE ($n = 12$), Wells ($n = 3$), Healthy ($n = 8$). **c** Box plots illustrating the normalized

modules scores in test biopsies. Each dot represents one test biopsy. LP ($n = 6$), AD ($n = 15$), PsO ($n = 8$), NeuD ($n = 5$), CLE ($n = 7$). **b**–**c** the central line of the box plot is the median. The box's edges are the lower (25th percentile) and upper quartiles (75th percentile). Whiskers extend to data points within 1.5 times the interquartile range (IQR). **d** UMAP projection showing clustering of test biopsies with sentinel biopsies having concordant disease diagnosis. Black and grey rings represent test and sentinel biopsies, respectively.

independent external cohort (called test group) demonstrated similar high diagnostic performance, both based on module dominance (Fig. 2c and Table 1) and unsupervised disease clustering (Fig. 2d).

We found that module dominance and unsupervised clustering persist across different anatomical locations of the disease. Biopsies obtained from classical body locations (trunk, extremities), as well as specialisites, such as palmoplantar and intertriginous areas, consistently

exhibited a dominant Th2 module in AD and Th17 module in PsO (Suppl. Fig. 5a). Furthermore, biopsies from LP lesions, whether from the skin or oral mucosa, consistently showed a dominant Th1 module (Suppl. Fig. 5a). Remarkably, within the same patient, biopsies not only clustered together regardless of the anatomical location (Suppl. Fig. 5b) but even if sampled months apart (Suppl. Figure 5c), indicating spatial and temporal stability of modules within a patient's disease.

**Table 1 | Performance metrics of module genes**

| | Sentinel probes | | | | Challenger probes | | | |
|---|---|---|---|---|---|---|---|---|
| Disease | Precision | Recall | Specificity | FM-Index | Precision | Recall | Specificity | FM-Index |
| PsO | 1.00 | 1.00 | 1.00 | - | 0.86 | 1.00 | 0.97 | - |
| AD | 0.94 | 1.00 | 0.99 | - | 1.00 | 0.93 | 1.00 | - |
| LP | 1.00 | 0.96 | 1.00 | - | 1.00 | 1.00 | 1.00 | - |
| CLE | 1.00 | 1.00 | 1.00 | - | 1.00 | 1.00 | 1.00 | - |
| ND | 1.00 | 1.00 | 1.00 | - | 1.00 | 0.86 | 1.00 | - |
| All | - | - | - | 0.95 | - | - | - | 0.9 |

Hierarchical clustering was used to calculate the diagnostic performance of sentinel patients and UMAP back-projection onto sentinel samples was used for challenger patients.

## Module profiles can classify other inflammatory skin diseases

Having demonstrated that immune modules can serve as diagnostic tool for model inflammatory skin diseases, we sought to explore their performance in diagnosing other less-well-characterized inflammatory diseases. We performed expression profiling of skin biopsies from patients with bullous pemphigoid (BP, $n = 17$), and maculopapular drug hypersensitivity reactions (DHR, $n = 10$). Initially, we aimed at identifying additional immune modules by performing differential gene expression analysis in these diseases against all sentinel biopsies used to defined immune models. While we did not find any differentially expressed genes (DEGs) capable of defining additional functional immune modules (Suppl. Fig. 6), we observed that the genes from the 7 modules were highly effective in clustering the disease samples in all three conditions, as evidenced by both heatmap (Fig. 3a) and UMAP analyses (Fig. 3b). Once again, the clustering efficiency per disease, assessed via the Fowlkes-Mallows (FM) index, was significantly higher when utilizing module genes (FM index of 0.85) compared to either the complete gene panel (FM index of 0.58) or a minimal gene set obtained through NS Forest (FM index of 0.82) (Fig. 3c). Notably, when comparing module expression, we observed unique combinations of modules expressed in a co-dominant (normalized expression >0.5 and within 30% of the highest expressed modules) specific to each disease: codominant Th2 and myeloid (macrophagic, neutrophilic, and eosinophilic) modules for BP; and codominant Th2, myeloid, and type I IFN modules for DHR (Fig. 3d). In summary, immune profiling and assessment of module expression represent powerful diagnostic tools and enable the development of a molecular immune cartography of inflammatory skin diseases.

## Module profiles provide efficient diagnostic support for erythroderma and undetermined rashes

We then proceeded to explore the potential utility of immune modules in cases of inflammatory skin diseases with ambiguous diagnoses, where conventional clinical and histology examinations may not yield conclusive results. Erythroderma exemplifies such cases, as the initial clinical presentation and histopathological analysis often fail to differentiate between psoriasis, atopic dermatitis, drug eruptions, and cutaneous lymphoma as the underlying causes. To address this challenge, we studied 30 erythroderma cases at our clinic (Suppl. Table 2), of which 12 remained uncertain following initial clinico-pathological assessments. After the exclusion of cutaneous lymphoma as an underlying cause through molecular assessment of blood and skin, we profiled the 30 erythroderma cases and projected the data onto our molecular cartography containing sentinel diseases. We found unequivocal clustering of erythroderma samples with psoriasis ($n = 10$), AD ($n = 16$), or drug eruptions ($n = 4$) (Fig. 4a) with the expression of their respective dominant module profile (Fig. 4b). Notably, we observed a perfect match (FM-index of 1.00, and a mean squared error of 0%) between the molecular diagnosis based on clustering and expression of dominant modules and the final diagnosis of the erythroderma established months after the initial presentation, relying on the clinical disease course, laboratory results and

drug testing. In contrast, the diagnostic performance of initial clinical and histological examination was significantly lower (FM-index of 0.89 for histology) with a large degree of uncertainty in their prediction.

We also conducted an analysis of 21 undetermined rashes, where diagnosis posed a challenge due to the absence or overlap of classical clinical and histopathological criteria (Suppl. Table 3). Molecular profiling of undetermined rashes and projecting the results onto our molecular reference cartography allowed a clear diagnosis of AD in 8 patients, as evidenced by unsupervised clustering with AD sentinels and the dominant expression of the Th2 module (Fig. 4c and Suppl. Table 3). Additionally, psoriasis was diagnosed in 10 patients through unsupervised clustering with psoriasis sentinels and the dominant expression of the Th17 module, LP was identified in 2 patients based on unsupervised clustering with Th1 sentinels, and the predominant expression of the Th1 module, and BP was identified in 1 patient who clustered with BP sentinels having co-dominant Th2, macrophagic, and eosinophilic modules (Fig. 4d). While the module-based diagnosis cannot be confirmed in all patients, therapeutic targeting of the dominant immune module in 6 patients led to complete clearance of the rash (Suppl. Table 3), supporting the functional relevance of the modules and the accuracy of the diagnosis. Hence, immune profiling, involving clustering against a set of well-defined sentinel probes and determining the dominant immune module, demonstrates diagnostic for both erythrodermas and undetermined skin rashes.

## Module matching to treatment target can increase response rates in treatment-naïve patients and in non-responders

Because immune modules can be directly targeted therapeutically, we next investigated whether matching the dominant module in the biopsy to its targeted therapy could improve the response to therapy in a cohort of 80 patients. Patients received treatment with anti-IL-4RA or anti-IL-13 (anti-Th2), anti-IL23 or anti-IL-17A (Anti-Th17), or JAK1 inhibitors (with the ability to target Th1) based on their diagnosis of AD, psoriasis, or LP established by clinical and histopathology criteria. Responses to therapy were assessed at week 16, defining responders as those with EASI > 75, PASI > 75, and LPSI > 75, and non-responders as those with EASI < 30, PASI < 30, and LPSI < 30. Profiling of the pre-treatment biopsy revealed 46 dominant Th2, 33 dominant Th17, and 1 dominant Th1 profile (Fig. 5a). All responding patients ($n = 60$) displayed a matched profile, while non-responding patients ($n = 19$) had both matched and non-matched profiles (Fig. 5a), indicating that profile matching is a prerequisite but not a guarantee for mounting a therapeutic response. Our data also indicate that profile matching in pre-treatment biopsies can increase responses to targeted therapy by eliminating non-matched patients. In fact, in our cohort of 80 patients, therapeutic responses would have increased from 76% to 83% if therapy was selected based on module matching.

We further profiled 17 post-treatment biopsies from non-responding patients undergoing targeted treatment with anti-Th2 ($n = 11$) or anti-Th17 ($n = 6$) therapies (Table 2). Among these 17 patients, 14 (82%) displayed a mismatched profile, while 3 (18%) were matched despite being non-responders. Six of the 14 patients with mismatched

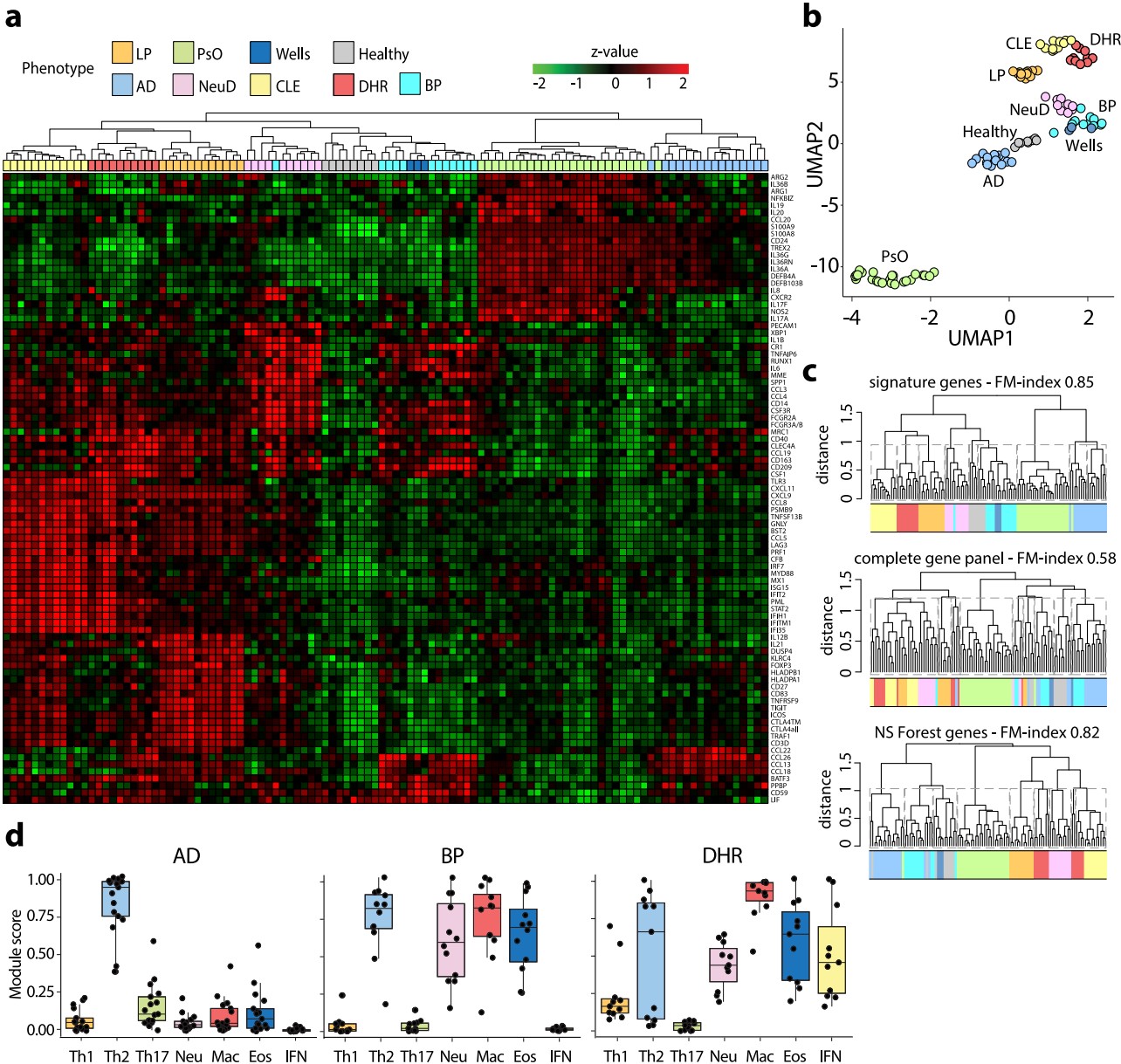

**Fig. 3 | Immune modules can classify non-sentinel inflammatory skin diseases such as bullous pemphigoid and maculopapular drug-hypersensitivity reactions. a** Heatmap displaying module profiles of maculopapular drug-hypersensitivity reaction (DHR) (red) and bullous pemphigoid (BP) (turquoise) plotted against the established module-based cartography of sentinel probes. **b** UMAP projection of sentinel, DHR, and BP probes showing disease clustering based on module expression. **c** Hierarchical clustering of DHR and BP probes based on module genes, the complete gene panel, or the NS Forest minimal genes, shown as dendrogram. Clustering accuracy is indicated by the Fowlkes–Mallows (FM) index. **d** Box plots illustrating immune module scores in DHR ($n = 10$), and BP ($n = 12$) compared to AD. Each dot represents one probe. The central line of the box plot is the median. The box's edges are the lower (25th percentile) and upper quartiles (75th percentile). Whiskers extend to data points within 1.5 times the interquartile range (IQR).

immune profiles were rematched to receive the corresponding targeted treatment (anti-IL4R for dominant Th2 modules, anti-IL-17A for dominant Th17 modules, and JAK inhibitors for dominant Th1), and all 6 patients achieved complete therapeutic responses (Table 2). Representative clinical and histological images of non-responding patients and their responses after rematch are provided in Suppl. Fig. 7. Altogether, these data suggest that mismatches between the dominant module and treatment target are likely the cause for the lack of therapeutic responses, and rematching can efficiently reverse this.

Interestingly, pre-treatment biopsies were available in 5 non-responding AD patients, and profiling revealed an initially matched module with the correct therapeutic targeting of the dominant Th2 module (Fig. 5b). However, despite a decrease in the Th2 module

expression under anti-IL-4RA therapy, we observed the emergence of a dominant Th1 module in all 5 non-responding skin lesions. Once again, patients achieved complete therapeutic responses by switching to module-guided treatment to JAK inhibitors targeting Th1, suggesting that the observed module switch to Th1 was the cause of the anti-IL-4RA therapy resistance. For the remaining 9 mismatched non-responding patients, we are unable to determine whether the mismatch is also consequence of a therapy-induced module switch or whether it results from a wrong targeting of a pre-existing mismatched module, because we did not collect pre-treatment biopsies.

In conclusion, our data suggest that module identification and therapeutic matching in patients' biopsies prior to the start of immune-

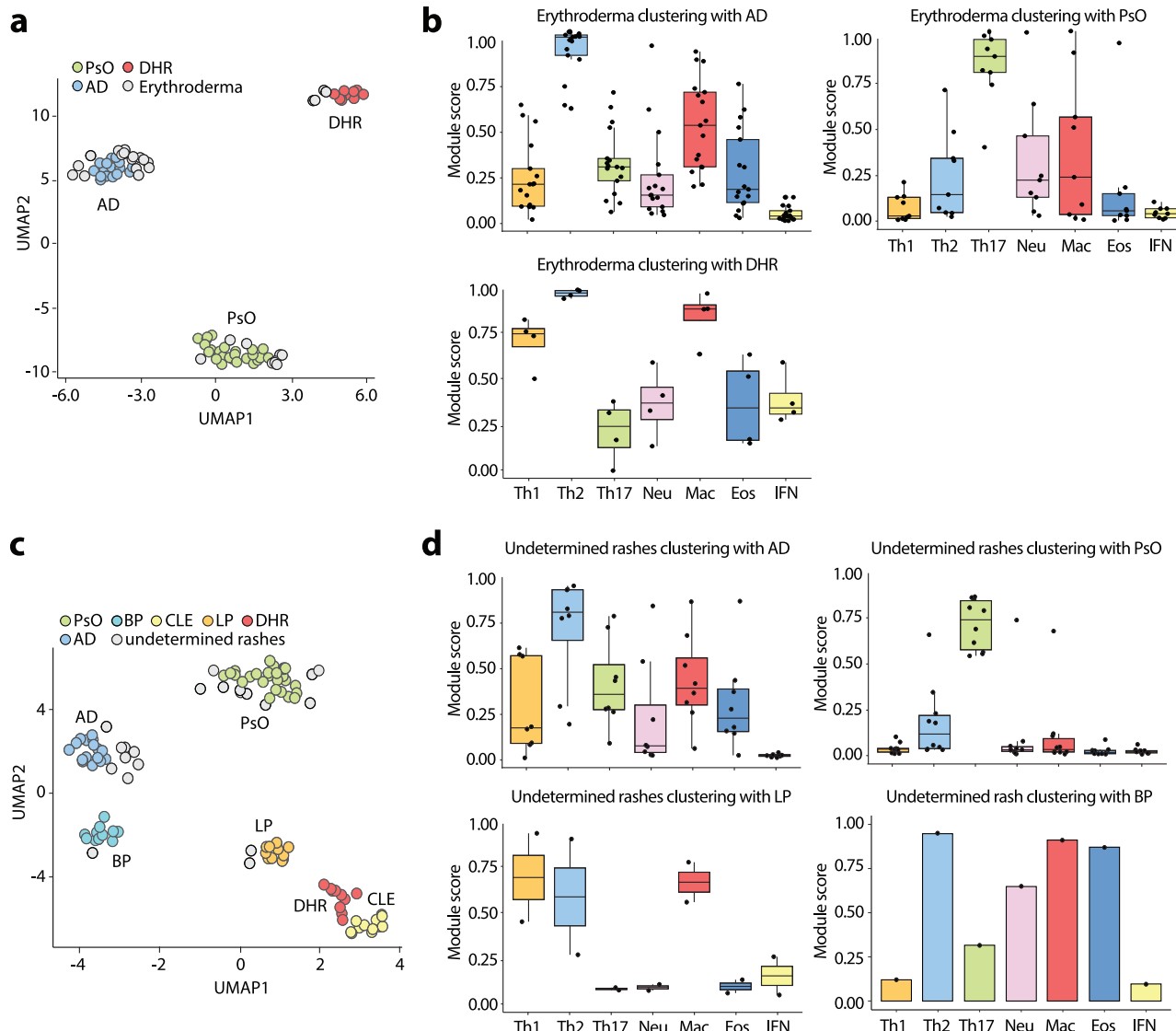

**Fig. 4 | Erythroderma and undetermined skin rashes display module profiles of AD, PsO, LP, and DHR sentinels. a** Module-based UMAP projection of erythroderma probes (grey circles) plotted against the established psoriasis (green circles), AD (blue circles), and DHR (red circles) cartography, showing disease-specific clustering. **b** Box plots showing immune module expression scores in erythroderma probes split by the sentinel disease (AD, $n = 16$; PsO, $n = 10$; or DHR, $n = 4$) they cluster with. Each dot represents one probe. **c** Module-based UMAP projection of undetermined rashes (grey circles) plotted against the established

PsO (green circles), AD (blue circles), CLE (yellow circles), BP (turquoise circles), and DHR (red circles) cartography, showing disease-specific clustering. **d** Box plots showing immune module expression scores in erythroderma probes split by the sentinel disease (AD, $n = 8$; PsO, $n = 10$; LP, $n = 2$; or BP, $n = 1$) they cluster with. **b** and **d**, the central line of the box plot is the median. The box's edges are the lower (25th percentile) and upper quartiles (75th percentile). Whiskers extend to data points within 1.5 times the interquartile range (IQR).

targeted treatment can improve clinical response rates and can be used to understand and guide therapeutic choices in non-responders.

## Discussion

Despite significant advances in the field of targeted therapies for inflammatory skin diseases, there remains a notable gap in molecular assessment that could guide diagnosis and personalized treatment decisions. In our study, we aimed to address this gap by employing innovative transcriptomic profiling of skin biopsies. This involved cross-comparing model inflammatory skin diseases to identify core immune modules involved in their pathogenesis. The identified transcriptional modules demonstrated a remarkable capacity in categorizing inflammatory skin diseases, offering powerful diagnostic support for clinical practice. Importantly, molecular profiling of module genes also facilitated the unequivocal classification of undetermined and erythrodermic cases, where clinico-pathological assessment alone was

inconclusive. Furthermore, accurate matching of the dominant modules present in pre-treatment biopsies with the immunological pathway targeted by the patient's treatment was found to be essential for eliciting a therapeutic response. This highlights the utility of the module-based approach in enhancing patients' likelihood of responding to treatment by avoiding obligatory failures associated with mismatched cases. Additionally, the identification of the dominant module in skin samples from non-responders provided unprecedented insights into the mechanisms underlying disease progression and efficiently guided targeted treatment choices through rematching the dominant module with the correct drug target.

While the concept of precision medicine was originally established in oncology[37], its application to chronic inflammatory diseases is still in its infancy[38,39]. Attempts to define biomarkers associated with treatment response have shown promising leads, such as the association of HLA-Cw6 in psoriasis with responses to anti-IL-12/23 but not

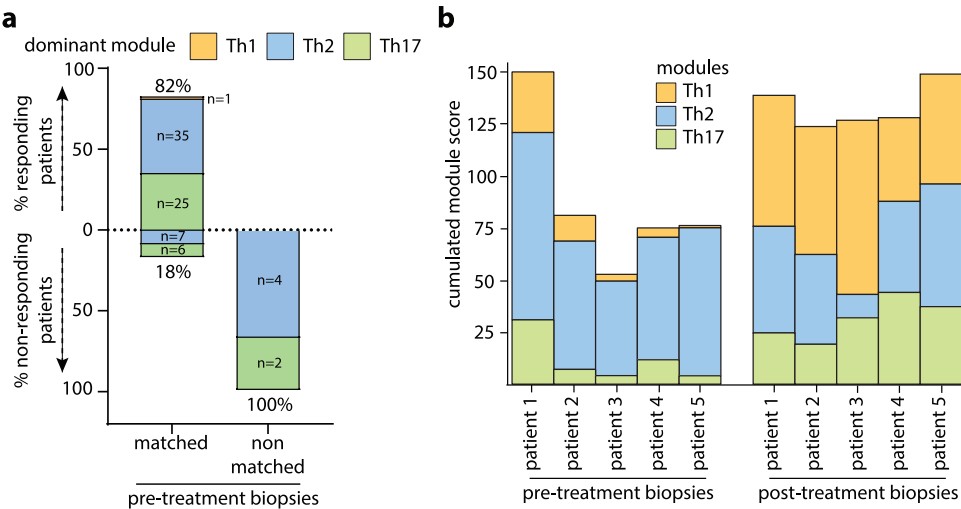

**Fig. 5 | Module matching to treatment targets to enhance therapeutic efficacy. a** Profiling of 80 pre-treatment biopsies from patients with AD, PsO, and LP undergoing targeted therapy, separated into responders or non-responders based on clinical scoring at week 16. The dominant expression module in skin probes (Th1, Th2 or Th17) was matched to the treatment target, defined as Th2 for anti-IL4R and IL-13, Th17 for anti-IL-23 and IL-17A/F, and Th1 for JAK1/2 inhibitors. Data are represented as percentages of responding and non-responding patients in the matched and non-matched groups. **b** Cumulative Th1, Th2, and Th17 module scores in pre- and post-treatment biopsies in 5 non-responding AD patients undergoing anti-Th2 treatment with anti-IL4R.

**Table 2 | Treatments and module matching of non-responding patients**

| Patient ID | Age | Gender | Initial Clinical Dx | Treatment | Module scores | | | Module matching | Re-matched treatment - Response |
|---|---|---|---|---|---|---|---|---|---|
| | | | | | Th1 | Th2 | Th17 | | |
| NR_001 | 67 | M | AD | Dupilumab | **0.82** | 0.11 | 0.32 | non matched | - |
| NR_002 | 66 | M | AD | Dupilumab | **0.61** | 0.5 | 0.25 | non matched | - |
| NR_003 | 66 | M | AD | Dupilumab | **0.66** | 0.29 | 0.52 | non matched | - |
| NR_004 | 79 | M | AD | Dupilumab | **0.6** | 0.42 | 0.19 | non matched | Baricitinib – 100% response |
| NR_005 | 33 | M | AD | Dupilumab | **0.43** | 0.15 | 0.03 | non matched | Upadacitinib- 90% response |
| NR_006 | 88 | F | PsO | Tildrakizumab | 0.41 | **0.95** | 0.23 | non matched | Dupilumab – 90% response |
| NR_007 | 65 | M | PsO | Tildrakizumab | 0.36 | **0.88** | 0.38 | non matched | Dupilumab – 100% response |
| NR_008 | 66 | M | PsO | Ixekizumab | 0.05 | **0.84** | 0.33 | non matched | Dupilumab – 90% response |
| NR_009 | 61 | M | AD | Dupilumab | 0.51 | **0.58** | 0.37 | matched | - |
| NR_010 | 59 | M | AD | Tralokinumab | 0.25 | **0.76** | 0.13 | matched | - |
| NR_011 | 58 | M | PsO | Secukinumab | 0.1 | **0.98** | 0.1 | non matched | - |
| NR_012 | 56 | M | PsO | Secukinumab | 0.32 | **0.96** | 0.12 | non matched | - |
| NR_013 | 65 | M | PsO | Ixekizumab | 0.19 | **0.97** | 0.47 | non matched | - |
| NR_014 | 47 | F | AD | Dupilumab | 0.17 | 0.1 | **0.38** | non matched | - |
| NR_015 | 55 | M | AD | Dupilumab | 0.19 | 0.3 | **0.55** | non matched | - |
| NR_016 | 59 | M | AD | Dupilumab | 0.1 | 0.11 | **0.36** | non matched | Ixekizumab – 90% response |
| NR_017 | 18 | F | PsO | Guselkumab | 0.04 | 0.08 | **0.72** | matched | - |

Bold scores correspond to the dominant module score for each patient.

anti-IL-17A[40–42]. Other studies using bulk transcriptomics[35,43], single-cell RNA sequencing[44], or spatial transcriptomics[45] have identified distinct gene signatures capable of categorizing only psoriasis and atopic dermatitis, but not other diseases. Moreover, the signatures did not provide direct links with the treatment choice as they did not involve functional cytokine pathways targetted by therapies. Our study now presents a robust precision medicine approach for inflammatory skin diseases, not only categorizing a broad range of diseases but also offering individualized guidance for selecting the right treatment.

The clinical relevance of our module-based profiling approach underscores the urgent need for its integration into clinical practice to effectively manage patients with inflammatory skin diseases, particularly those with unclear diagnosis or prior to introducing targeted treatments. Scaling up this implementation necessitates possibly automated processes compared to the current time-consuming method. Additionally, optimizing tissue procurement techniques through minimally invasive skin techniques will be needed to maximize patient compliance.

Module-based profiling revealed unique mechanisms underlying resistance to targeted therapies. In fact, dominant Th2 profiles in pre-treatment biopsies of atopic dermatitis patients can transition into dominant Th1 profiles associated with a non-responsive state to anti-Th2 therapies. The clinical significance of this transition is evidenced by the prompt therapeutic response to anti-Th1 therapies with JAK inhibitors. These findings suggest a finely tuned immune balance that can be disrupted by targeted therapies, like the phenomenon observed in paradoxical psoriasis where anti-TNF therapies induce an upregulation of the type I IFN pathway[46]. Thus, our approach represents a valuable tool to detect immune shifts, offering mechanistic insights into clinical failures, and guiding a rational choice for more effective

therapies. Future studies will have to identify biomarkers capable of predicting the development of such immune shifts. This is likely to require a broader approach extending beyond measuring the transcription of 600 immune genes and involving multi-omics technologies. A limitation of our module-based approach is its reduced discriminatory power for certain immune modules, such as the eosinophilic module, due to the minimal differences in gene expression between Wells disease, healthy skin, and other diseases. Expanding gene expression analysis beyond the current 600 immune genes could also help identify additional genes that enhance the classification capability of these modules.

Module-based profiling also holds promise in identifying the dominant immune module in poorly characterized inflammatory diseases, thereby facilitating the repurposing of available therapies. For instance, our study has revealed a predominant Th2 module in bullous pemphigoid, aligning with recent data on the efficacy of anti-IL-4RA therapy[47]. Furthermore, this profiling approach enables the identification of subdominant module expression, contributing to the molecular definition of disease endotypes. Our findings demonstrate varying degrees of subdominant co-expression of Th1 and Th17 modules alongside the dominant Th2 pathway in atopic dermatitis, as well as a subtle yet discernible co-expression of neutrophilic and IFN modules with the dominant Th17 module in psoriasis, as previously reported[48,49]. Nevertheless, the functional relevance of these additional pathways in relation to responsiveness to targeted therapies and the development of immune shifts remains to be elucidated.

## Methods

### Patients and Biopsies

The study was approved by the institutional review board of the Lausanne University Hospital CHUV Switzerland and the local ethics committee (Commission cantonale d'éthique de la recherche sur l'être humain, CER-VD), in accordance with the Helsinki Declaration and were reviewed by the ethical committee board of the canton of Vaud, Switzerland. 264 patients with inflammatory skin diseases undergoing routine diagnostic skin biopsies at the Department of Dermatology in Lausanne were included in the study (Suppl. Table 4). Frozen biopsies from consented patients were deposited into the Swiss Biobanking Platform (SBP)-accredited Dermatology biobank before analysis. The median age of patients was 58 (range 19–95) and 36% were women (Suppl. Table 5). We included biopsies of 12 common inflammatory skin diseases from untreated patients with psoriasis ($n = 58$), AD ($n = 66$), LP ($n = 18$), cutaneous LE ($n = 19$), Wells syndrome ($n = 3$), bullous pemphigoid ($n = 17$), drug hypersensitivity reaction ($n = 10$), hidradenitis suppurativa ($n = 5$), pyoderma gangrenosum ($n = 5$), Sweet syndrome ($n = 2$), and dissecting cellulitis ($n = 3$). To construct the molecular cartography, we selected patients for sentinel biopsies according to the following criteria: psoriasis patients having chronic plaque-type psoriasis, diagnosed clinically according to standard morphological criteria and confirmed by histopathology; atopic dermatitis patients with clinically-well defined disease, confirmed by histopathology and history of atopy with elevated blood IgE levels; lichen planus patients having classical cutaneous lichen planus diagnosed clinically, confirmed by histopathology with presence of civatte bodies on direct immunofluorescence; cutaneous lupus erythematosus patients having either discoid or subacute cutaneous LE, diagnosed clinically and confirmed by histopathology, with presence of a lupus band on direct immunofluorescence; patients with Sweet syndrome, hidradenitis suppurativa, pyoderma gangrenosum, dissecans cellulitis (all neutrophilic dermatoses) as well as wells syndrome, selected according to classical clinical criteria and confirmed by histopathology; bullous pemphigoid patients

diagnosed clinically and confirmed by histopathology with positive direct immunofluorescence and serological testing for anti-BP180/230; drug hypersensitivity reactions (DHR) patients presenting with maculo-papular eruptions and having a history of specific drug intake, confirmed by histopathology and in-vivo skin tests or ex-vivo blood testing for drug hypersensitivity.

In addition, patients with unclear clinical and pathological diagnosis were investigated, including 30 erythroderma patients and 21 patients with undetermined rashes. Histological slides (hematoxylin and eosin stain) of the erythroderma patients were blindly reviewed by three independent dermato-histopathologists. They provided a percentage of the probability for the diagnosis (choosing from the 3 entities – eczema, psoriasis, drug hypersensitivity reaction) based on the histological criteria observed. For studies on therapy guidance, pre-treatment biopsies of additional 80 patients and post-treatment biopsies of additional 17 non-responding patients were included. Biopsy location included trunk (32.7%), extremities (51.3%), palmoplantar (4.8%), intertriginous (2.2%), head and neck (7.4%), oral (1.1%), and genital (0.4%). Patient ethnicity was the following: 83% Caucasian, 4% Middle Eastern, 7% Hispanic, 3% African, and 2% Asian, 1% other ethnicities.

### Sample processing and gene expression analysis

Skin biopsies were immediately snap-frozen in liquid nitrogen and stored at −80 °C until processing. RNA was isolated using the TRIzol/chloroform method and a tissue homogenizer (Thermo Fisher Scientific). Quality control was run on a Fragment analyzer (Agilent) to select RNAs with A260/A280 value of ≥1.7, RNA integrity >9 and a DV300 > 50%. mRNA expression of 600 immune targets was analysed with the nCounter Human Immunology V2 panel, including 20 customized probes (Nanostring Technologies, Seattle, WA, USA) on the nCounter platform (Nanostring Technologies) using 100 ng of RNA per skin sample. This commercial panel was extensively validated in-house for accuracy, repeatability, and reproducibility before analysing the study samples. A quality check was run for each sample before including it into the analysis. The raw data produced by nCounter were normalized to remove technical variability and to enhance biological differences. First, an individual normalization factor was calculated by dividing the geometric mean of the raw gene expression of positive probes of each biopsy by the mean value of all geometric means. The raw expression of each biopsy was then multiplied by its own normalization factor, giving an intermediate expression level. Then, a second normalization factor was calculated as above but using the housekeeping genes probes. The previous intermediate expression levels were multiplied by this normalization factor to obtain the final normalized expression levels. The normalized expression levels were finally transformed by converting all values below 1 to 1 and by applying a logarithmic transformation on base 2. The distribution profiles of the raw counts, normalized counts, and scaled data are depicted in Suppl. Fig. 8.

For RNAseq, RNA from skin biopsies ($n = 36$ PsO, $n = 24$ AD, $n = 22$ Lichen planus, $n = 11$ cutaneous lupus, $n = 11$ pyoderma gangrenosum) was isolated using the QIAzol Lysis Reagent (Qiagen) and miRNeasy Mini Kit (Qiagen) according to manufacturer's protocol. RNASeq libraries were generated using the TruSeq Stranded Total RNA Kit (Illumina) according to manufacturer's high sample protocol. Finally, samples were sequenced on an Illumina HiSeq4000 as paired-end with a read length of 2×150 bp and an average output of 40 Mio reads per sample and end. Sequence alignment was performed using STAR aligner with human genome reference hg38. RNAseq count data was normalized and then transformed using variance stabilizing transformation from the Bioconductor package DESeq2.

### Module definition and diagnostic cartography

To determine the list of genes corresponding to each inflammatory module, a differential gene expression analysis was performed

between each sentinel disease group and all the other sentinels using Limma{limma} based on linear models. Adjusted *p* values using Benjamini-Hochberg correction was used. The false-discovery rate threshold was set to 0.01. The algorithm was used on normalized expression levels of the sentinel biopsies. Modules were defined as the group of genes having a normalized expression fold change above 2 ($\log_2$ FC = 1), with *p* value below 0.01 (-$\log_{10}$ *P* = 2). The gene list was further optimized by eliminating genes that were already present in other modules and those that functionally were not fitting into the identified pathway. For comparison, a minimal set of genes was also identified using the NS-Forest algorithm. The algorithm was used on the normalized expression levels of the sentinel biopsies. The code is available on a repository using the following packages python 3.8.10, scanpy 1.9.6, anndata 0.9.2, and NS-Forest 3.9.2 (https://github.com/JCVenterInstitute/NSForest). Clustering of samples was then performed with either the total list of genes (~600), the module genes only, or the NS-Forest genes using the R functions umap{umap} and hclust{stats}. The agglomeration method "complete" was used in hclust and the distance was calculated with the normalized expression levels using the Pearson correlation. Heatmaps were generated with pheatmap{pheatmap} using the z-scores of module genes.

### Diagnostic validation and performance
A threshold was set for each module as the local minimum of the bimodal distribution using sentinel samples that are positive and negative for the module. A logit transform was then used on the aggregated scaled expression to define an activation score equal to 0.5 when the expression is equal to the threshold, the activation score being 0 when the module is not expressed and 1 when it reaches maximal expression. This logit transformation leads to a more interpretable score value between 0 and 1. To validate the utility of modules genes in classifying samples based on diagnostic, test biopsies from other patients with sentinel diseases were used. Clustering performance using the different gene sets was measured with sensitivity, recall, precision, and the Fowlkes-Mallows Index (square root of the product between recall and precision) using the R functions ml_test{mltest} and FM-index{dendextend}.

### Patient treatment and module alignment
To profile pre-treatment biopsies, we enrolled consecutive consenting patients and presenting at our clinic between January 2020 and December 2023, qualifying for systemic treatment, and undergoing diagnostic biopsies within 3 weeks of initiating systemic treatment. This cohort comprised 39 individuals with atopic dermatitis, initially presenting with an EASI score > 22.2, who received standard Dupilumab dosing (300 mg Q2W). Additionally, we included 6 patients with bullous pemphigoid, starting with an initial BPDAI score > 51, who were administered off-label standard Dupilumab dosing. Among the participants were 30 with plaque psoriasis, exhibiting a PASI > 75, receiving standard dosing of guselkumab (*n* = 6), secukinumab (*n* = 4). ixekizumab (*n* = 5), risankizumab (*n* = 6), tildrakizumab (*n* = 5), bimekizumab (*n* = 1), and ustekinumab (*n* = 3). Furthermore, one patient with severe LP was treated with JAK1/2 inhibitor baricitinib at a dosing of 4 mg once daily.

All patients were evaluated at week 16 of treatment. Responders were defined as those achieving at least a 75% improvement in their respective disease score, whereas non-responders were those not meeting a 30% amelioration. A retrospective analysis of the pre-treatment biopsy was conducted in both groups (responders and non-responders) to identify the dominant module and match it to the treatment target. Matching criteria were as follows: a dominant Th2 module for patients treated with Dupilumab and Tralokinumab, a dominant Th17 module for patients with anti-IL-17A (Secukinumab and Ixekizumab) or anti-IL-23 (Tildrakizumab and Guselkumab), and a dominant Th1 module for patients receiving JAK1/2 inhibitor Baricitinib.

Consenting non-responding patients to targeted treatment (*n* = 17) treated at our clinic between January 2020 and June 2024, underwent biopsy of the non-responding lesions followed by profile matching with the treatment target. In *n* = 6 patients, treatment was adjusted based on the biopsy profile, and clinical response was reassessed after 16 weeks of therapy.

### Reporting summary
Further information on research design is available in the Nature Portfolio Reporting Summary linked to this article.

## Data availability
Nanostring transcriptomics datasets generated for this publication and re-analyzed from Di Domizio J. et al.[36] are deposited at GEO data repository under the accession number GSE280220 and GSE193068 respectively. All other data are available in the article and its Supplementary files or from the corresponding author upon request. Source data are provided with this paper.

## Code availability
The codes needed to reproduce analysis and findings of this study are available in the GitHub repository https://github.com/derchuv/persomed.

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

## Acknowledgements

We thank I. Surbeck, A. Darbellay, and A. Joncic for technical assistance. This work was funded by the Swiss National Science Foundation (310030B_182834, 310030_204835, 4078P0_198470) and the Leenaards Foundation to MG.

## Author contributions

T.S., J.D.D., and A.G. produced the data. T.S. coordinated patient recruitment and sampling. J.D.D and A.G generated nanostring transcriptomics data and associated bioinformatics analysis. A.Y., R.J., F.Me., F.S., C.S., S.B., F.Mi., M.L., H.W., C.P., and C.C. selected patients, performed skin biopsies, and analyses. N.G.S, S.E., and K.E. provided RNA sequencing data. M.V., D.H., and E.G. performed histopathological analyses. R.G. supervised the statistics and bioinformatics analyses. M.G. conceived and supervised the work, and wrote the manuscript along with T.S., J.D.D., and A.G. as well as comments from co-authors.

## Competing interests

The authors declare no competing interests.
