## [Peer Review file · Nature Communications]

Immune Modules to guide Diagnosis and Personalized Treatment of Inflammatory Skin Diseases

Corresponding Author: Professor Michel Gilliet

Version 0:

Reviewer comments:

Reviewer #1

(Remarks to the Author)

Caplanusi et al describe gene expression modules that differentiate inflammatory skin diseases such as psoriasis, atopic dermatitis and lichen planus. They then show that these modules can be used to facilitate the diagnosis of "difficult" cases and guide disease treatment.

This is an interesting study and its conclusions are supported by the authors' findings. At the same time, the manuscript could be further improved by providing a few clarifications and explicitly acknowledging some limitations.

Line 106: the authors mention skin manifestation of severe Covid-19, but it's not clear how the relevant transcription profiles were generated. Were they retrieved from the paper by Domizio et al? If so, can the authors provide the dataset identifier, clarify what platform was used (RNA-seq or Nanostring) and how many cases were analysed.

Line 113: On a related note, the authors describe a dominant macrophage signature in the skin manifestations of severe Covid-19. However, the data supporting this statement is not shown in any of the Figures. This should be addressed.

Figure 2b/4d: Are the differences reported in the Figures statistically significant? Also, the statement that the type I IFN module is co-dominant in DHR does not seem to match the data in Figure 4d.

Methods: Were the patients from the sentinel and challenger cohorts receiving treatment at the time of the biopsy? If so, how was this potential confounder taken into account?

Methods/Discussion: No information is provided about patient ethnicity. If the examined cases were mostly of European ancestry, this should be explicitly acknowledged as a limitation of the study.

Discussion: Liu et al also developed a gene-expression based classifier to aid the diagnosis of ambiguous skin rashes (RashX; <https://pubmed.ncbi.nlm.nih.gov/35427179/>). In what respect is the authors' approach an improvement? Did the expression modules they identified overlap with those reported by Liu et al?

Minor:

Given the Methods are at the end of the manuscript, it would be helpful to provide some key information in the main text. This includes the criteria used to generate the expression modules, the number of patients in the replication cohort, the number of patients with BP and DHR.

It would be helpful to explain to the non-expert reader what are the main features of Wells syndrome and why it was included in the analysis.

Figure 3 and Tables 2-5 could be moved to the Supplementary Materials

Lines 230-232: The authors mention 12 non responders and state that 9 had AD and 2 psoriasis. That leaves one case unaccounted for.

(Remarks on code availability)

Reviewer #2

(Remarks to the Author)

Immune Modules to guide Diagnosis and Personalized Treatment of Inflammatory Skin Diseases
NCOMMS-24-24396-T

Dear Editors and Authors,

It has been a pleasure to read the manuscript.

The manuscript is based on results from Nanostring gene expression analyses on skin biopsies from patients with different inflammatory skin diseases. The gene expression profiles comprise 600 immune-related genes. Furthermore, RNA seq data on a smaller cohort is included in the first part of the manuscript. The differentially expressed genes are suggested to be useful in routine clinical setting in addition to current clinical practice. There are presented data for using the gene expression profiles across 2 platforms, different anatomical areas and time points.

The manuscript represents a great amount of work and is presented in five sections.

The manuscript interesting and an important contribution to the field.

Please find my comments below.

Comments

The skin biopsies are collected from patients with different inflammatory skin diseases that include psoriasis, atopic dermatitis, lichen planus, cutaneous LE, neutrophilic diseases, erythrodermic patients.

While the authors write that these are well-defined clinical cases used as “sentinels” in the manuscript, the tables and text contain no information about the subtypes of these clinical entities that are equally well-described and should be included.

The authors mention multiple genes that are differentially expressed in the groups of inflammatory skin diseases (from line 87-116). It will give the readers a better sense of the differences in the gene expression that is the foundation for this manuscript if the supplementary tables include the raw gene expression counts, normalized gene expression, and box plots for the genes mentioned from line 87-116. It will also give the readers an overview of the differences in the gene expression that is the foundation for this manuscript.

It is unclear to me why the authors have added cases from “Wells syndrome and skin manifestations of severe COVID” in this manuscript, and the authors explanation would be useful in the manuscript.

Comments on differential expression analyses and the identification of “modules” (which could be called “gene groups”, see comments for Nomenclature).

The figure 1 text reads “ $P < 0.05$.” while the Methods section reads “Modules were defined as the group of genes having a fold change above 2 ($\log_2 FC = 1$), a p value below 0.01 ($-\log_{10} P = 1.5$) and playing a significant role in the inflammatory pathway of each sentinel disease.”

The manuscript contains differential expression analyses on one disease group compared to the rest (Figure 1). Did the authors choose the most differentially expressed genes in all analyses? There seem to be other genes in the volcano plots that are differentially expressed but not named – are these included in the gene groups (modules)?

Why did the authors choose targeted method and a p-value cut-off of 0.01 instead of using some of the correction/adjustment methods available in Limma, so you get an adjusted p-value concerning, for example, false-discovery rate?

Line 147: The authors state that module dominance and unbiased clustering persist across different anatomical locations of the disease. What does “unbiased clustering” mean in this context? The clustering is based on targeted analyses of genes that previously have been chosen among those that are differentially expressed.

Figure 3

The UMAP plot depicts a group of samples from healthy donors. It is interesting that this group clusters so closely with the Wells, even in this targeted heat map, given the infiltrated lesion often observed in these patients.

In Figure 2b, on the other hand, it has high “module” scores for “eosino”, “macro” and “neuro”, while healthy have low scores for these genes.

What could be the explanation for this?

Validation cohort for modules

Line 144: The external validation cohort (including disease severity and disease subtypes) does not seem to be described (or has not been found by this reviewer).

3) Module profiles can classify other inflammatory skin diseases

The graphs on Figure 4d seem to show that both BP and DHR have a mixed immune profile.

The UMAP on 4b includes these two groups and the inflammatory skin disease that clusters furthest away -AD. This might be interpreted a bit selectively portrayed.

What is the diagnostic value of this – the clinical presentation of BP and DHR are quite dissimilar.

Part 4: Module profiles provide efficient diagnostic support for erythroderma and 180 undetermined rashes.

The authors have included 30 patients with erythroderma which is a sizable cohort.

Erythrodermic patients are in some cases challenging to treat and patient history is important in these cases. It is relevant to exclude lymphoma in the included patients as the authors have done.

Although I think that this is an important contribution, I am not able to understand Figure 4 b. Are the erythrodermic patients included in the box plots with the rest of the group it clusters with? Could the plots indicate the different groups?

Table 2: What does the percentage represent in the histopathology column?

Initial clinical diagnosis: there seem to be DHR in patients with no suspected drugs?

PRP is mentioned in the table but not in the manuscript text as a differential diagnosis.

5) Module matching to treatment target can increase response rates in treatment-naïve patients and in non-responders.

The authors analyse the transcriptional profile of 80 patients with AD, PSO and LP that are undergoing treatment.

These include 60 patients responding to treatment with expected gene expression profiles, and 20 nonresponding patients of which some had expected gene expression profiles.

New transcriptional analysis is carried out of 12 patients out of 20 nonresponders during treatment. These include 9 patients with AD and 2 psoriasis which equals 11 patients (not 12?). Are new biopsies collected during treatment?

Of the 12 patients, 3 gene expression analyses confirmed the gene expression of the expected group, while 9 did not. Could this be a consequence of treatment? The description in this paragraph is hard to follow and should be accompanied by a flowchart and an overview of gene expression results for patients before and after treatment and disease subtype. Ultimately, 7 of the 9 biopsies with a gene expression profile that belonged to another group responded to a treatment change. The authors should explain why 7/9 patient represent "all patients" that have complete response.

Furthermore, can the results be attributed disease subtypes? The cohort includes psoriasis, AD and LP, which in the first part of the manuscript result in clear gene expression group changes. What are the explanations for nonmatching results in another cohort of the same group of patients? Could there be a challenge in terms of capturing the full spectrum of immune responses specific to each condition?

Nomenclature

The nomenclature should be clearer – for example gene groups instead of modules. Validation - or test group, instead of 'challenger group', which is common nomenclature to define the data you use to validate your prediction models.

Graphs

The authors should have separate supplementary tables with raw data, normalized data, source data for each box plot. And box plots for each gene mentioned in Part 1.

Discussion

The discussion could be more in-depth and mention for example limitations to this study.

(Remarks on code availability)

Reviewer #3

(Remarks to the Author)

In this manuscript, the authors demonstrated that specific gene modules derived from Nanostring profiling of model inflammatory skin diseases showed diagnostic efficacy of ambiguous inflammatory skin conditions and improving treatment response in non-responders to targeted therapies. This study provides some interesting aspects for an usefulness of gene module-based molecular diagnosis and exploring treatment targets. This reviewer has some comments.

Major

- It is somewhat unclear how the authors selected patients for sentinel biopsies although there is some information on the method section. As sentinel biopsy samples are critical to generate gene modules, it would be noteworthy to provide diagnostic and selection criteria for those samples. Also, it would be more informative if the authors provide some representative clinical and histological pictures for each model diseases.

- It is unclear how the authors select module genes from each model disease (Figure 1 volcano plots). Are there any cutoff values for gene expression fold change and/or adjusted p-values? It seems that the selected genes in the modules are selected somewhat in a biased manner based on the previously known disease immunopathogenesis. Furthermore, each module has a different number of genes which might affect the enrichment efficiency of each module. What is the effect of the number of genes in each module? What if the authors use the same cutoff values and/or number of genes in each module to differentiate the diseases?

- Where are Sweet syndrome samples in UMAP of NeuD on UMAP of supplementary figure 1? (I can only see DC, HS, PG samples)

- Where are results for skin manifestations of severe COVID samples to drive macrophage gene module?

- In Figure 4, it would be more informative to see all disease UMAP to understand the location of BP and DHR in relation to

other model diseases. Furthermore, although BP samples were clustered with Well's syndrome, Eos module score was lesser than that of Th2 and Mac. What are the driving molecules for clustering between BP and Well's syndrome? In addition, did BP and DHR samples not have differentially expressed genes compared to model diseases or there differentially expressed genes were not able to differentiate them from other diseases?

- Figure 5: What is the underlying history for skin diseases in erythroderma patients? In certain cases, erythrodermic patients have prior skin conditions which are useful to characterize their erythrodermic status. In addition, DHR type erythroderma patients showed a highly increased Th2 gene module. Is there any possibility that those patients are drug reaction with eosinophilia and systemic symptoms (DRESS)?

- Figure 6b: Were 12 patients new nonresponding to targeted treatments or some patients from Figure 6a? What are the representative clinical and pathological photos for non-responders?

Minor

- Line 51: anti-IL4-R -> anti-IL-4RA?

(Remarks on code availability)

Version 1:

Reviewer comments:

Reviewer #1

(Remarks to the Author)

The authors have addressed all the issues I raised.

The answer describing the differences with RashX was clear and informative, so I would suggest that the Authors incorporate it in the Discussion.

(Remarks on code availability)

Reviewer #3

(Remarks to the Author)

In response to this reviewer's comments, the authors well revised the manuscript accordingly.

Especially, the representative clinical and histological pictures provide convincing information.

The performance and usefulness of modules will require continuous validation in other clinical settings.

(Remarks on code availability)

REVIEWER COMMENTS

Reviewer #1 (Remarks to the Author):

Caplanusi et al describe gene expression modules that differentiate inflammatory skin diseases such as psoriasis, atopic dermatitis and lichen planus. They then show that these modules can be used to facilitate the diagnosis of “difficult” cases and guide disease treatment.

This is an interesting study and its conclusions are supported by the authors' findings. At the same time, the manuscript could be further improved by providing a few clarifications and explicitly acknowledging some limitations.

We thank the reviewer for his positive feedback. Below we have addressed all the points raised.

Line 106: the authors mention skin manifestation of severe Covid-19, but it's not clear how the relevant transcription profiles were generated. Were they retrieved from the paper by Domizio et al? If so, can the authors provide the dataset identifier, clarify what platform was used (RNA-seq or Nanostring) and how many cases were analyzed.

Indeed, the profiles were retrieved from the manuscript Domizio et al, Nature 2022. We have now added a new Supplementary Figure 3 to show how the myeloid signature was dissected into neutrophilic, macrophagic, and eosinophilic sub-signatures based on the comparison of neutrophilic diseases, COVID skin lesions, and Wells syndrome. In the Supplementary Figure 3 we also illustrate the number of cases analyzed. Analysis was done by the same Nanostring Technology. The link to the dataset in GEO will be given with the final manuscript,

Line 113: On a related note, the authors describe a dominant macrophage signature in the skin manifestations of severe Covid-19. However, the data supporting this statement is not shown in any of the Figures. This should be addressed.

See above, data are now included as New Supplementary Figure 3.

Figure 2b/4d: Are the differences reported in the Figures statistically significant?

We now added new Supplementary Table 1 to show statistical significance between dominant modules and all other modules expressed by sentinel diseases.

Also, the statement that the type I IFN module is co-dominant in DHR does not seem to match the data in Figure 4d. Address

We have now improved the text, stating that in BP there is co-dominance of the Th2 and myeloid modules (neutrophilic, macrophagic, and eosinophilic), whereas DHR display co-dominance of the Th2, myeloid, and the IFN modules (new text, lines 179-180).

Methods: Were the patients from the sentinel and challenger cohorts receiving treatment at the time of the biopsy? If so, how was this potential confounder taken into account?

None of the patients in the sentinel or challenger group were under therapy.

Methods/Discussion: No information is provided about patient ethnicity. If the examined cases were mostly of European ancestry, this should be explicitly acknowledged as a limitation of the study.

We are now providing the ethnicity distribution of patients included in the study: 83% were Caucasian, 4% Middle Eastern, 7% Hispanic, 3% African, and 2% Asian, 1% other ethnicities. The information was included in the New Material & Methods section, lines 386-387).

Discussion: Liu et al also developed a gene-expression based classifier to aid the diagnosis of ambiguous skin rashes (RashX; <https://pubmed.ncbi.nlm.nih.gov/35427179/>). In what respect is the authors' approach an improvement? Did the expression modules they identified overlap with those reported by Liu et al?

The following points are the main differences and improvements compared to the Liu et al. study:

1. Liu et. Al identified signatures on scRNA seq samples that allow differentiation only between AD and PsO. They also tested lichen planus and bullous pemphigoid but could not find specific signatures, but rather found clustering with psoriasis and AD, respectively. By contrast we provide a comprehensive diagnostic cartography that allows identification of 8 or

more different inflammatory skin diseases (PsO, AD, LP, CLE, Wells syndrome, neutrophilic diseases, bullous pemphigoid, drug hypersensitivity reactions). As shown in New Figure 4C, we can classify undetermined cases into PsO, AD, LP, BP, CLE, and DHR.

2. Liu et al. found that DEG (>100 genes) within the TRM cells of psoriasis and AD provide a signature that allow to distinguish psoriasis and AD. These genes are not linked to Th17 and Th2. In fact, the authors mention that the TRM subtype (Th2 or Th17) is not able to segregate the disease. By contrast we provide clinically relevant signature linked to the activated immune pathway and show their relevance not only for diagnosis of the disease but also to choose the correct targeted treatment modality.

Minor:

Given the Methods are at the end of the manuscript, it would be helpful to provide some key information in the main text. This includes the criteria used to generate the expression modules, the number of patients in the replication cohort, the number of patients with BP and DHR.

Modules were defined as the group of genes having a normalized expression fold change above 2 ($\log_2 FC=1$), with p value below 0.01 ($-\log_{10} P = 2$). The gene list was further optimized by eliminating genes that were already present in other modules and those that functionally were not fitting into the identified pathway. We included this explanation into the Material & Methods section, lines 426-429. Regarding the number of patients with BP and DHR, we included this information in the results section, lines 166-167.

It would be helpful to explain to the non-expert reader what are the main features of Wells syndrome and why it was included in the analysis.

Wells disease, also known as eosinophilic cellulitis, is a rare inflammatory condition characterized by pathogenic eosinophil infiltration of the dermis. We included this information into the results section, lines 108-110.

Figure 3 and Tables 2-5 could be moved to the Supplementary Materials

As suggested by the reviewer, the data were moved to new Supplementary Figure 5 and new Supplementary Tables 2-5

Lines 230-232: The authors mention 12 non responders and state that 9 had AD and 2 psoriasis. That leaves one case unaccounted for.

We apologize for the mistake. In accordance with the comments of reviewer 3, we eliminated Figure 5B and are now providing a comprehensive Table, new Table 2, describing a total of 17 non responders, 10 with AD and 7 with PsO.

Reviewer #2 (Remarks to the Author):

Immune Modules to guide Diagnosis and Personalized Treatment of Inflammatory Skin Diseases
NCOMMS-24-24396-T

Dear Editors and Authors,

It has been a pleasure to read the manuscript.

The manuscript is based on results from Nanostring gene expression analyses on skin biopsies from patients with different inflammatory skin diseases. The gene expression profiles comprise 600 immune-related genes. Furthermore, RNA seq data on a smaller cohort is included in the first part of the manuscript. The differentially expressed genes are suggested to be useful in routine clinical setting in addition to current clinical practice. There are presented data for using the gene expression profiles across 2 platforms, different anatomical areas and time points.

The manuscript represents a great amount of work and is presented in five sections.

The manuscript interesting and an important contribution to the field.

Please find my comments below.

We thank the reviewer for finding our manuscript interesting and important. Below we have addressed all the comments.

COMMENTS:

The skin biopsies are collected from patients with different inflammatory skin diseases that include psoriasis, atopic dermatitis, lichen planus, cutaneous LE, neutrophilic diseases, erythrodermic patients. While the authors write that these are well-defined clinical cases used as “sentinels” in the manuscript, the tables and text contain no information about the subtypes of these clinical entities that are equally well-described and should be included.

We thank the reviewer for pointing this out. We have now included the following information about the well-defined sentinels into the Material and Methods section, lines 360-382:

To construct the molecular cartography, we selected patients for sentinel biopsies according to the following criteria: psoriasis patients having chronic plaque-type psoriasis, diagnosed clinically according to standard morphological criteria and confirmed by histopathology; atopic dermatitis patients with clinically-well defined disease, confirmed by histopathology and history of atopy with elevated blood IgE levels; lichen planus patients having classical cutaneous lichen planus diagnosed clinically, confirmed by histopathology with presence of civatte bodies on direct immunofluorescence; cutaneous lupus erythematosus patients having either discoid or subacute cutaneous LE, diagnosed clinically and confirmed by histopathology, with presence of a lupus band on direct immunofluorescence; patients with Sweet syndrome, hidradenitis suppurativa, pyoderma gangrenosum, dissecans cellulitis (all neutrophilic dermatoses) as well as wells syndrome, selected according to classical clinical criteria and confirmed by histopathology; bullous pemphigoid patients diagnosed clinically and confirmed by histopathology with positive direct immunofluorescence and serological testing for anti-BP180/230; drug hypersensitivity reactions (DHR) patients presenting with maculo-papular eruptions and having a history of specific drug intake, confirmed by histopathology and in-vivo skin tests or ex-vivo blood testing for drug hypersensitivity.

The authors mention multiple genes that are differentially expressed in the groups of inflammatory skin diseases (from line 87-116). It will give the readers a better sense of the differences in the gene expression that is the foundation for this manuscript if the supplementary tables include the raw gene expression counts, normalized gene expression, and box plots for the genes mentioned from line 87-116. It will also give the readers an overview of the differences in the gene expression that is the foundation for this manuscript.

As requested, we now added new Supplementary Figure 8 to show box plots of the raw gene counts, the normalized gene expression, and the scaled gene expression. The different counts are also provided as source data.

It is unclear to me why the authors have added cases from “Wells syndrome and skin manifestations of severe COVID” in this manuscript, and the authors explanation would be useful in the manuscript.

We added Wells syndrome and skin manifestations of severe COVID to dissect the myeloid signature into neutrophilic, macrophagic, and eosinophilic sub-signatures. These data are now shown in new Supplementary Figure 3.

Comments on differential expression analyses and the identification of “modules” (which could be called “gene groups”, see comments for Nomenclature).

We prefer to keep the term module to indicate the immune signatures but have added a clear statement that these modules are gene groups (result section, line 122).

The figure 1 text reads “P<0.05.” while the Methods section reads “Modules were defined as the group of genes having a fold change above 2 (log₂ FC=1), a p value below 0.01 (-log₁₀ P = 1.5) and playing a significant role in the inflammatory pathway of each sentinel disease.”

We apologize for the mistake and corrected as p value below 0.01.

The manuscript contains differential expression analyses on one disease group compared to the rest (Figure 1). Did the authors choose the most differentially expressed genes in all analyses? There seem to be other genes in the volcano plots that are differentially expressed but not named – are these included in the gene groups (modules)?

As discussed in the response to comments to reviewer 1, modules were defined as the group of genes having a normalized expression fold change above 2 (log₂ FC=1), with p value below 0.01 (-

log₁₀ P = 2). The gene list was further optimized by eliminating genes that were already present in other modules and those that functionally were not fitting into the identified pathway. We included this explanation into the Material & Methods section, lines 426-429.

Why did the authors choose targeted method and a p-value cut-off of 0.01 instead of using some of the correction/adjustment methods available in Limma, so you get an adjusted p-value concerning, for example, false-discovery rate?

Indeed, the adjusted p-values using Benjamini-Hochberg correction was used. The false-discovery rate threshold was set to 0.01. This information was added Material & Methods section, lines 423-424.

Line 147: The authors state that module dominance and unbiased clustering persist across different anatomical locations of the disease. What does “unbiased clustering” mean in this context? The clustering is based on targeted analyses of genes that previously have been chosen among those that are differentially expressed.

We thank the reviewer for pointing at this, we agree that this clustering is not unbiased as it relies on selected module genes, the correct term is unsupervised clustering. We have changed the terminology throughout the manuscript.

Figure 3

The UMAP plot depicts a group of samples from healthy donors. It is interesting that this group clusters so closely with the Wells, even in this targeted heat map, given the infiltrated lesion often observed in these patients.

In Figure 2b, on the other hand, it has high “module” scores for “eosino”, “macro” and “neutro”, while healthy have low scores for these genes.

What could be the explanation for this?

The UMAP constructs a low-dimensional representation of the data by taking into consideration all the sentinel samples and the presence or absence of the module genes. Because Wells and healthy skin samples both have low expression levels of most of module genes compared with other sentinel diseases, they tend to cluster close to each other. This is why we need to use multiple analyses to get a correct immune profile including Heatmap for individual gene expression, Euclidian distance based on Pearson correlation for clustering group, module score for the activation of immune pathways, and UMAP for assessing the similarity in immune profiles. Moreover, the low number of Wells samples (n=3) might reduce the weight of this immune profile when considering clustering calculation in the UMAP.

Validation cohort for modules

Line 144: The external validation cohort (including disease severity and disease subtypes) does not seem to be described (or has not been found by this reviewer).

In Supplementary Table 4, we provide the comprehensive list of samples with disease information. The selection criteria of the validation cohort was the same as for the sentinels.

3) Module profiles can classify other inflammatory skin diseases

The graphs on Figure 4d seem to show that both BP and DHR have a mixed immune profile. The UMAP on 4b includes these two groups and the inflammatory skin disease that clusters furthest away -AD. This might be interpreted a bit selectively portrayed.

What is the diagnostic value of this – the clinical presentation of BP and DHR are quite dissimilar.

We thank the referee for pointing this out. We have now changed the UMAP in new Figure 3b where we included all sentinel diseases (Psoriasis, Lichen planus and Lupus erythematosus) along with BP and DHR to reflect the real clinical-relevant comparisons. Also, in the unclear cases in new Figure 4c, we have added such examples to illustrate the power of our approach.

Part 4: Module profiles provide efficient diagnostic support for erythroderma and 18 undetermined rashes.

The authors have included 30 patients with erythroderma which is a sizable cohort.

Erythrodermic patients are in some cases challenging to treat and patient history is important in these cases. It is relevant to exclude lymphoma in the included patients as the authors have done.

Although I think that this is an important contribution, I am not able to understand Figure 4 b.

Are the erythrodermic patients included in the box plots with the rest of the group it clusters with? Could the plots indicate the different groups?

We corrected the labels in New Figure 4b-d to make it clearer that only erythrodermic or undetermined rashes are plotted in the box plots.

Table 2: What does the percentage represent in the histopathology column?

The histological slides (hematoxylin and eosin stain) of the erythroderma patients were blindly reviewed by three independent dermato-histopathologists with several years of experience. They provided a percentage of the probability for the diagnosis (choosing from the 3 entities – eczema, psoriasis, drug hypersensitivity reaction) based on the histological criteria observed. We have now included this explanation in the Material & Methods Section, lines 378 – 282.

Initial clinical diagnosis: there seem to be DHR in patients with no suspected drugs?

There was a clinical suspicion, although no clear history of drug intake.

PRP is mentioned in the table but not in the manuscript text as a differential diagnosis.

We removed this from the list of differential diagnosis in Supplementary Table 1

5) Module matching to treatment target can increase response rates in treatment-naïve patients and in non-responders.

The authors analyse the transcriptional profile of 80 patients with AD, PSO and LP that are undergoing treatment.

These include 60 patients responding to treatment with expected gene expression profiles, and 20 nonresponding patients of which some had expected gene expression profiles.

New transcriptional analysis is carried out of 12 patients out of 20 nonresponders during treatment. These include 9 patients with AD and 2 psoriasis which equals 11 patients (not 12?).

We apologize for the mistake. According to comments of the reviewers we are now providing a detailed table with information on 17 patients, including 10 AD and 7 psoriasis. (New Table 2)

Are new biopsies collected during treatment?

The post-treatment biopsies of non-responding patients are indeed harvested during treatment.

Of the 12 patients, 3 gene expression analyses confirmed the gene expression of the expected group, while 9 did not. Could this be a consequence of treatment? The description in this paragraph is hard to follow and should be accompanied by a flowchart and an overview of gene expression results for patients before and after treatment and disease subtype.

As mentioned above we are now providing a detailed table with information on 17 patients (5 new patients derived from an extended recruitment period until July 2024), including 10 AD and 7 psoriasis. (New Table 2). The point about the mismatch as consequence of the targeted treatment is well taken. We have pre-treatment biopsied in only 5 patients (shown in Figure 5b) and the data show a correct diagnosis with matched therapeutic treatment. The mismatch in the non-responding post-treatment biopsy was therefore consequence of a switch in the inflammatory module under therapy. For the remaining 9 patients, we do not have pre-treatment biopsies and can therefore not make any conclusion whether the mismatch results from a switch or whether it was already present prior therapy. We have added a comment in the Result Section, lines 259-262.

Ultimately, 7 of the 9 biopsies with a gene expression profile that belonged to another group responded to a treatment change. The authors should explain why 7/9 patient represent “all patients” that have complete response.

To make this figure clearer and as suggested by another reviewer, we included a table giving the characteristics of the molecular expression and therapeutic matching and re-matching (new Table 2)

Furthermore, can the results be attributed disease subtypes? The cohort includes psoriasis, AD and LP, which in the first part of the manuscript result in clear gene expression group changes. What are the explanations for nonmatching results in another cohort of the same group of patients? Could there be a challenge in terms of capturing the full spectrum of immune responses specific to each condition

We think there can be 2 reasons for mismatches:

- wrong initial diagnosis or concomitant second disease (e.g. AD in a psoriasis patient)
- immune switch under targeted treatment and paradoxical reactions

In Figure 5b, we show that for 5 patients with a correct diagnosis and matched therapeutic treatment based on the analysis of pre-treatment biopsies switched their molecular phenotype under therapy. For the remaining 9 mismatched patients, we do not have pre-treatment biopsies. We are therefore unable to conclude whether a switch or a mismatch was already present prior therapy. We have added a comment in the Result Section, lines 259-262.

Nomenclature

The nomenclature should be clearer – for example gene groups instead of modules. Validation - or test group, instead of 'challenger group', which is common nomenclature to define the data you use to validate your prediction models.

We added the new label “test group” instead of “challenger group” throughout the manuscript.

Graphs

The authors should have separate supplementary tables with raw data, normalized data, source data for each box plot.

And box plots for each gene mentioned in Part 1.

We added new Supplementary Figure 8 to show box plots of the raw gene counts, the normalized gene expression, and the scaled gene expression. The different counts are also provided as source data.

Discussion

The discussion could be more in-depth and mention for example limitations to this study.

The discussion provides the following discussion about limitations of the described approach:

- 1) Need for scaling up possibly through automated processes compared to the current time-consuming method. Additionally, optimizing tissue procurement techniques through minimally invasive skin techniques will be needed to maximize patient compliance (Discussion, lines 307-310)
- 2) The use of limited number of genes within the modules does not provide enough data granularity for further analysis including identification of predictive markers for immune shifts observed in the study. This is likely to require a broader approach extending beyond measuring transcription of 600 immune genes and involving multi-omics technologies. We have commented on this in the New Discussion, lines 320-323.

Reviewer #3 (Remarks to the Author):

In this manuscript, the authors demonstrated that specific gene modules derived from Nanostring profiling of model inflammatory skin diseases showed diagnostic efficacy of ambiguous inflammatory skin conditions and improving treatment response in non-responders to targeted therapies. This study provides some interesting aspects for an usefulness of gene module-based molecular diagnosis and exploring treatment targets. This reviewer has some comments.

Major

- It is somewhat unclear how the authors selected patients for sentinel biopsies although there is some information on the method section. As sentinel biopsy samples are critical to generate gene modules, it would be noteworthy to provide diagnostic and selection criteria for those samples. Also, it would be more informative if the authors provide some representative clinical and histological pictures for each model diseases.

We have now included the following information about the well-defined sentinels into the Material and Methods section, lines 360-382:

To construct the molecular cartography, we selected patients for sentinel biopsies according to the following criteria: psoriasis patients having chronic plaque-type psoriasis, diagnosed clinically according to standard morphological criteria and confirmed by histopathology; atopic dermatitis patients with clinically-well defined disease, confirmed by histopathology and history of atopy with elevated blood IgE levels; lichen planus patients having classical cutaneous lichen planus diagnosed clinically, confirmed by histopathology with presence of civatte bodies on direct immunofluorescence; cutaneous lupus erythematosus patients having either discoid or subacute cutaneous LE, diagnosed clinically and confirmed by histopathology, with presence of a lupus band on direct immunofluorescence; patients

with Sweet syndrome, hidradenitis suppurativa, pyoderma gangrenosum, dissecans cellulitis (all neutrophilic dermatoses) as well as wells syndrome, selected according to classical clinical criteria and confirmed by histopathology; bullous pemphigoid patients diagnosed clinically and confirmed by histopathology with positive direct immunofluorescence and serological testing for anti-BP180/230; drug hypersensitivity reactions (DHR) patients presenting with maculo-papular eruptions and having a history of specific drug intake, confirmed by histopathology and in-vivo skin tests or ex-vivo blood testing for drug hypersensitivity.

As requested, we also added clinical and histological images for each model disease in new Supplementary Figure 1 and 7

- It is unclear how the authors select module genes from each model disease (Figure 1 volcano plots). Are there any cutoff values for gene expression fold change and/or adjusted p-values? It seems that the selected genes in the modules are selected somewhat in a biased manner based on the previously known disease immunopathogenesis.

As discussed in the response to comments to reviewer 1, modules were defined as the group of genes having a normalized expression fold change above 2 ($\log_2 FC=1$), with p value below 0.01 ($-\log_{10} P = 2$). The gene list was further optimized by eliminating genes that were already present in other modules and those that functionally were not fitting into the identified pathway. We included this explanation into the Material & Methods section, lines 426-429.

Furthermore, each module has a different number of genes which might affect the enrichment efficiency of each module. What is the effect of the number of genes in each module?

Because module scores calculation depends on the average of expression levels of all the genes present in a module, module scores with a low number of genes in a module (e.g. 4 genes for Th2 and Eosino modules) tend to be more variable. Indeed, the weight per gene is higher in modules with a low number of genes than a high number of genes. This leads to the close clustering of samples (heatmap, UMAP) that are positive only for these low-genes modules including Healthy skin, AD, and Wells samples. However, by adding up the different analyses, our tool is powerful enough to get the correct immune profiles and to distinguish these diseases.

What if the authors use the same cutoff values and/or number of genes in each module to differentiate the diseases?

The cutoff values were the same for each module and the biased selection of genes relevant to the pathway did not decrease clustering efficiency per disease.

- Where are Sweet syndrome samples in UMAP of NeuD on UMAP of supplementary figure 1? (I can only see DC, HS, PG samples).

Sweet's syndrome samples are part of the test group and not sentinel group. In the UMAP of Figure 2d, all NeuD samples (sentinel and test) are present but labels are not shown to ease reading. However, for the reviewer only we provide a labeled UMAP in P2P Figure 1 below to show the location of Sweet samples.

P2P Figure 1. UMAP projection showing clustering of test probes with sentinel probes having concordant disease diagnosis. Black and grey rings represent test and sentinel probes, respectively. Diseases belonging to the group of neutrophilic dermatoses are color coded. HS, Hidradenitis suppurativa; PG, Pyoderma gangrenosum; DC, Dissecting cellulitis; Sweet, Sweet's syndrome.

- Where are results for skin manifestations of severe COVID samples to drive macrophage gene module?

We added new Supplementary Figure 3 to show the dissection of the myeloid signature into neutrophilic, macrophagic, and eosinophilic based on the analysis of neutrophilic diseases, COVID, and Wells.

- In Figure 4, it would be more informative to see all disease UMAP to understand the location of BP and DHR in relation to other model diseases.

We changed the UMAP in new Figure 3b where we included all sentinel diseases along with BP and DHR.

Furthermore, although BP samples were clustered with Well's syndrome, Eos module score was lesser than that of Th2 and Mac. What are the driving molecules for clustering between BP and Well's syndrome?

As discussed above, because clustering calculation depends on both the presence and absence of module genes, we need to use multiple analyses to get a correct immune profile including Heatmap for individual gene expression, Euclidian distance based on Pearson correlation for clustering group, module score for the activation of immune pathways, and UMAP for assessing the similarity in immune profiles. This is why BP and Wells may cluster together but having clearly different profiles of module scores.

In addition, did BP and DHR samples not have differentially expressed genes compared to model diseases or there differentially expressed genes were not able to differentiate them from other diseases?

We searched for DEGs that could be specific for BP and DHR compared to all sentinels. We show in new Supplementary Figure 6 that we could not find new signatures and that upregulated genes were those already present in the modules.

- Figure 5: What is the underlying history for skin diseases in erythroderma patients? In certain cases, erythrodermic patients have prior skin conditions which are useful to characterize their erythrodermic status. In addition, DHR type erythroderma patients showed a highly increased Th2 gene module. Is there any possibility that those patients are drug reaction with eosinophilia and systemic symptoms (DRESS)?

In our database we observed that DRESS samples have indeed a prominent Th2 module, however also 50% of maculopapular DHR also have prominent Th2 signatures (Figure 3d), indicating that the presence of the Th2 module is not specific for DRESS.

- Figure 6b: Were 12 patients new nonresponding to targeted treatments or some patients from Figure 6a?

Of the patients with non-responding post-treatment biopsies, only 5 had pre-treatment biopsies. These are the patients shown in Figure 5b.

What are the representative clinical and pathological photos for non-responders?

We now provide representative clinical and histopathology images of mismatches skin lesion and response to matched therapies in new Supplementary Figure 7.

Minor

- **Line 51: anti-IL4-R -> anti-IL-4RA?** We changed this accordingly.

Reviewer #1 (Remarks to the Author):

The authors have addressed all the issues I raised.

The answer describing the differences with RashX was clear and informative, so I would suggest that the Authors incorporate it in the Discussion.

We have improved our discussion to highlight the differences with previous studies including RashX that is referenced as ref.44. RashX only provides means to distinguish psoriasis from atopic dermatitis and does not provide direct links with the treatment choice as it involves gene signatures expressed by TRMs not involving functional cytokine pathways targeted by therapies. The discussion has been modified as follows:

Other studies using bulk transcriptomics (43 35), single-cell RNA sequencing (44) or spatial transcriptomics (45) have identified distinct gene signatures capable of categorizing only psoriasis and atopic dermatitis, but not other diseases. Moreover, the signatures did not provide direct links with the treatment choice as they did not involve functional cytokine pathways targeted by therapies. Our study now presents a robust precision medicine approach for inflammatory skin diseases, not only categorizing a broad range of diseases but also offering individualized guidance for selecting the right treatment.

Reviewer #2 comments mediated by reviewer #3

1• The concern regarding the selection of clinical cases for “sentinels” has been well addressed by additional information for patient selection and representative histopathology.

2• The concern regarding additional data requested for DEG (raw gene counts, normalized gene expression, scaled gene expression, and box plots) have been well presented.

3• The concern regarding Wells syndrome and skin manifestations of severe COVID has been addressed by providing new data in new Supplementary Figure 3.

4• The concern regarding “gene modules” vs “gene groups” has been addressed by the authors have added a statement to indicate that the immune signatures are part of gene groups.

5• The concern regarding the discrepancy between figure 1 text ($P < 0.05$) vs methods ($P < 0.01$) has been addressed by unifying the term $P < 0.01$.

6• The concern regarding the method to select gene list in each gene module has been addressed by more clearly providing the strategy although there is a potential bias by manual selection of gene list.

7• The concern regarding the use of adjusted P-value (e.g. FDR value) has been addressed by indicating that the authors used the adjusted P-values using Benjamin-Hochberg correction.

8• The concern regarding the term “unbiased clustering” has been addressed by use alternative term “unsupervised clustering”.

9• The concern regarding the discrepancy of the closeness between healthy and Well’s syndrome samples between UMAP (Figure 3) and Figure 2b would be somewhat addressed by the authors’ explanation (e.g. potentially less discriminating power of eosinophil, macrophage, and/or

neutrophil modules compared to other modules and low number of Wells samples). This reviewer feels that those weakness would be discussed in the discussion section accordingly to clarify the potential limitation of the current modular approaches.

10• The concern regarding the information of validation cohort has been provided by the authors.

11• The concern regarding the unclear separation between DHR and BP has been somewhat addressed by analyzing most “sentinel” samples together.

12• The concern regarding the classification of undetermined or erythrodermic rash patients has been addressed by only plotting those patients in box plots.

13• The concern regarding the meaning of “percentage” in the histopathology column has been clarified.

14• The concern regarding the initially diagnosed DHR patients with no suspected drug has been clarified.

15• The concern regarding the PRP as a differential diagnosis has been addressed by that the authors removed it from the list of DDx.

16• The concern regarding the mismatch of the treatment non-responding patients (initially 12) has been addressed as actual 17 (10 AD and 7 psoriasis) patients.

17• The concern regarding the potential changes in gene module expression in post-treatment samples has been clarified by the authors and there is a new explanation for this limitation in the Result section, lines 259-262.

18• The concern regarding the information about patients who showed treatment response after changing the therapeutic based on their molecular signatures has been provided in new Table 2.

19• The concern regarding the nonmatching results in another cohort of the sample group patients has been addressed and discussed.

20• The concern regarding the nomenclature has been revised.

21• The concern regarding the additional description for the limitation of current study has been well addressed.

All comments addressed.

One discussion point suggested (POINT #9).

We thank the reviewer for finding convincing our different replies, and we have now addressed the point to discuss by adding the following in the discussion section:

A limitation of our module-based approach is its reduced discriminatory power for certain immune modules, such as the eosinophilic module, due to the minimal differences in gene expression between Wells disease, healthy skin, and other diseases. Expanding gene expression analysis beyond the current 600 immune genes could also help identify additional genes that enhance the classification capability of these modules.

Reviewer #3 (Remarks to the Author):

***In response to this reviewer's comments, the authors well revised the manuscript accordingly.
Especially, the representative clinical and histological pictures provide convincing information.
The performance and usefulness of modules will require continuous validation in other clinical settings.***

We thank the reviewer for finding our revised manuscript improved.